# Concurrent circulation of dengue serotype 1, 2 and 3 among acute febrile patients in Cameroon

**Huguette Simo Tchetgna**[1]*, **Francine Sado Yousseu**[1,2], **Basile Kamgang**[1], **Armel Tedjou**[1,3], **Philip J. McCall**[4], **Charles S. Wondji**[1,4]

**1** Centre for Research in Infectious Diseases, Yaoundé, Cameroon, **2** University of Buéa, Buéa, Cameroon, **3** University of Yaoundé I, Yaoundé, Cameroon, **4** Liverpool School of Tropical Medicine, Liverpool, United Kingdom

* huguette.simo@crid-cam.net/

**Data Availability Statement:** The data generated in the manuscript are available in Genbank (https://www.ncbi.nlm.nih.gov/nuccore/). accession numbers MW970757 to MW979765 These are the

## Abstract

Acute febrile patients presenting at hospitals in Douala, Cameroon between July and December 2020, were screened for dengue infections using real time RT-PCR on fragments of the 5' and 3' UTR genomic regions. In total, 12.8% (41/320) of cases examined were positive for dengue. Dengue virus 3 (DENV-3) was the most common serotype found (68.3%), followed by DENV-2 (19.5%) and DENV-1 (4.9%). Co-infections of DENV-3 and DENV-2 were found in 3 cases. Jaundice and headache were the most frequent clinical signs associated with infection and 56% (23/41) of the cases were co-infections with malaria. Phylogenetic analysis of the envelope gene identified DENV-1 as belonging to genotype V, DENV-2 to genotype II and DENV-3 to genotype III. The simultaneous occurrence of three serotypes in Douala reveals dengue as a serious public health threat for Cameroon and highlights the need for further epidemiological studies in the major cities of this region.

## Author summary

Acute febrile patients presenting at hospitals in Douala, Cameroon between July and December 2020, were screened for dengue infections by Polymerase chain reaction. In total, 12.8% (41/320) of cases examined were infected by dengue virus. Dengue virus 3 (DENV-3) was the most common serotype found (68.3%), followed by DENV-2 (19.5%) and DENV-1 (4.9%). Co-infections of DENV-3 and DENV-2 were found in 3 cases. Jaundice and headache were the most frequent clinical signs associated with infection and 56% (23/41) of the cases were co-infections with malaria. The simultaneous occurrence of three serotypes in Douala reveals dengue as a serious public health threat for Cameroon and highlights the need for further epidemiological studies in the major cities of this region.

link to access the sequences produced in the current article. https://www.ncbi.nlm.nih.gov/nuccore/MW979763.1 https://www.ncbi.nlm.nih.gov/nuccore/MW979764.1 https://www.ncbi.nlm.nih.gov/nuccore/MW979761.1 https://www.ncbi.nlm.nih.gov/nuccore/MW979765.1 https://www.ncbi.nlm.nih.gov/nuccore/MW979760.1 https://www.ncbi.nlm.nih.gov/nuccore/MW979759.1 https://www.ncbi.nlm.nih.gov/nuccore/MW979758.1 https://www.ncbi.nlm.nih.gov/nuccore/MW979762.1 https://www.ncbi.nlm.nih.gov/nuccore/MW979757.1.

**Funding:** This study was supported by the Global Challenges Research Fund (GCRF) and the UK Research and Innovation through the Partnership for Increasing the Impact of Vector Control project (PIIVeC, https://www.piivec.org/). Project grant reference MR/P027873/1. The funders had no role in study design, data collection and analysis, decision to publish, or preparation of the manuscript.

**Competing interests:** The authors have declared that no competing interests exist.

# Background

Dengue is one of the most important mosquito-borne infections worldwide and its disease burden, one of the greatest in global health. The global incidence of dengue has increased by over 30 fold in recent decades, reaching today's level of approximately 400 million infections affecting half of the world's population in the 125 countries at risk of infection [1,2]. Dengue results from an infection with dengue virus (DENV) of the genus *Flavivirus* (*Flaviviridae* family). DENV genome is a positive sense, non-segmented, single stranded RNA of ~10.7kb which comprise an open reading frame encoding for three structural proteins and seven non-structural (NS) proteins in the order 5' C-preM/M-E-NS1-NS2a-NS2b-NS3-NS4a-NS4b-NS5 3' [3]. In fact, DENV is a complex of four antigenically and genetically distinct viruses, referred to as dengue serotypes 1 to 4 (DENV-1, DENV-2, DENV-3, and DENV-4) and each serotype can be further subdivided into multiple genotypes.

Dengue infections in humans are typically inapparent but can involve a range of clinical presentations from mild fever to the complications that can result in the potentially fatal severe Dengue [4,5]. A major determinant of disease severity is the timing and sequence of infection as infection with one virus serotype confers a lifelong homotypic immunity but subsequent re-infection with a different serotype increases the risk of severe disease [4]. DENV is primarily transmitted to humans by the mosquitoes of the *Aedes* genus, with *Aedes aegypti* the primary vector and *Ae. albopictus* the secondary vector. The last two decades have been marked by an unprecedented global expansion of the vectors rapidly followed by dengue, the result of global-scale human-driven changes such as massively increased urbanization, deforestation and ineffective vector control, and latterly climate change, all contributing to increase in vector transmission from human to human [2].

In Africa, dengue outbreaks have been recorded more frequently in many countries including Gabon, Angola, Mozambique, Kenya, Ethiopia, Sudan and Burkina Faso [6,7,8]. All four DENV serotypes have been widely reported across Africa and the co-circulation of two or more serotypes, first reported from Gabon in 2013 [8] is becoming more common, *e.g.* Kenya [7], Burkina Faso [6], Nigeria [9], Tanzania [10] raising fears that severe dengue will be seen more frequently in future outbreaks in Africa.

In Cameroon numerous studies over the past twenty years have reported seroprevalence rates in rural and urban areas in Cameroon, with no major outbreak notifications [11,12]. Cross-sectional studies in Yaoundé and Kribi have reported DENV-1 and DENV-2 [13,14] and a small scale study conducted in Douala in 2017 showed that dengue serotype 1 was responsible of 10% of the acute febrile diseases found in that city [14]. Both *Ae. aegypti* and *Ae. albopictus*, occur in Douala where *Ae. aegypti* is predominant in downtown and *Ae. albopictus* more common in suburban areas [15]. The Douala populations of both species can transmit DENV-2, and at high transmission efficiency for *Ae. aegypti* [16]. Therefore, our aim was to estimate the implication of dengue fever among outpatients seen in four public hospitals in the city and determine the main active genotypes and serotypes.

# Material and methods

## Ethics statement

The study was approved by the National Ethical Committee for Human Research in Cameroon (N˚2019/06/1165/CE/CNERSH/SP) and the institutional ethical committee of the Liverpool School of Tropical Medicine (Research Protocol 19–012). Permission to perform the study was obtained from each hospital and written informed consent was obtained from every

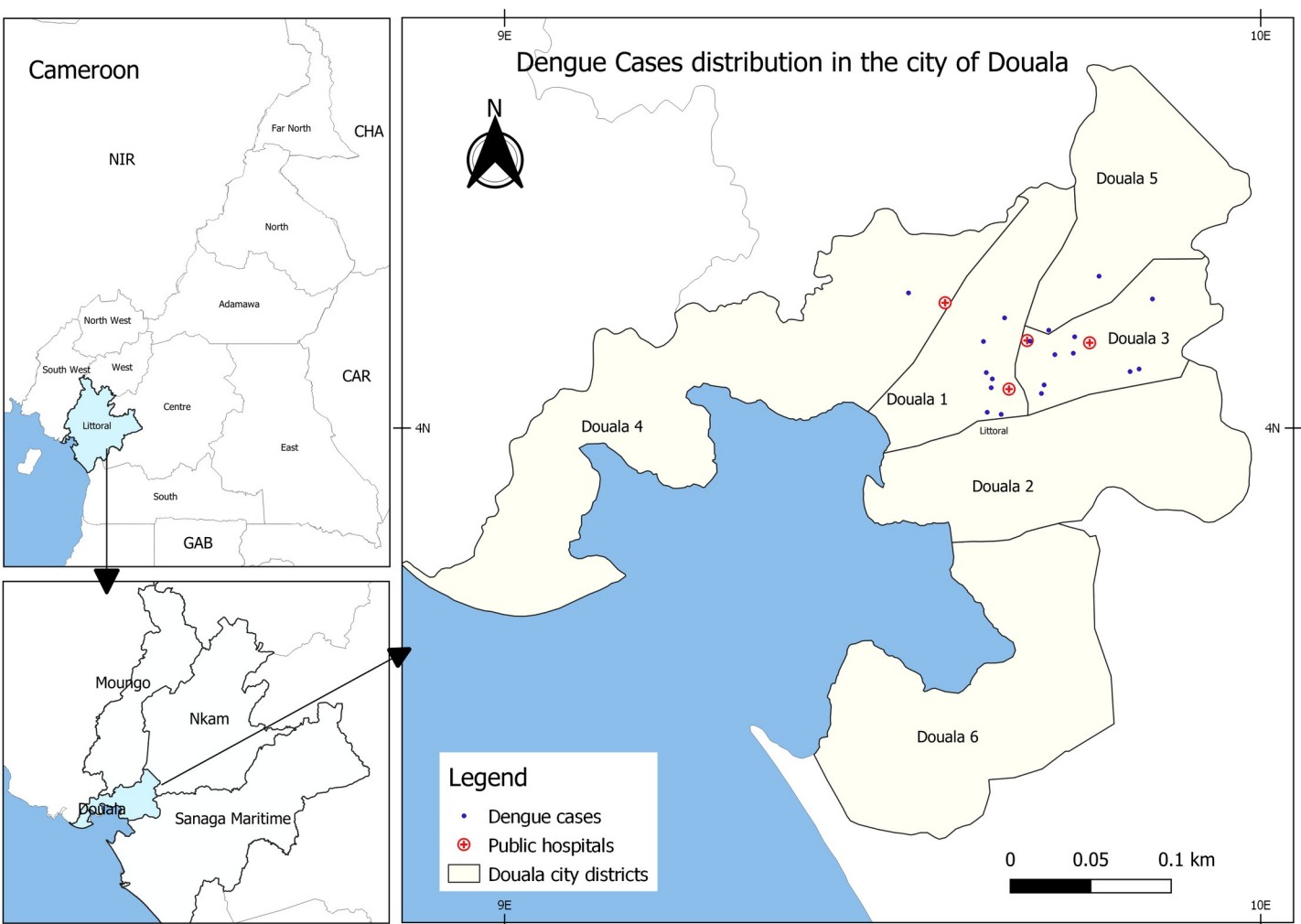

**Fig 1. Sampling sites and dengue cases distribution in the city of Douala.** The figure was built using the Cameroon shapefiles available at https://gadm.org/download_country_v3.html with QGIS Desktop V.3.14.16 software (https://www.qgis.org/en/site/).

patient who volunteered to be recruited. Assent and written consent for volunteers under 20 years old were obtained from their parents or legal guardian.

## Study design and patient inclusion

This cross-sectional study was conducted from July to December 2020 in 4 public hospitals in Douala (Fig 1), the largest city in Cameroon with approximately 3 million inhabitants. Douala has a warm and humid tropical climate with an average annual temperature and precipitation of 26˚C and 3600mm respectively. The port of Douala is the largest seaport of Central Africa, with significant international movement of goods and passengers.

Outpatients above 3 years old presenting with an acute febrile syndrome were enrolled in the study. The inclusion criteria required a fever (an axillary temperature above 38˚C) of less than 7 days duration plus at least three of the following symptoms: headache, asthenia, myalgia, rash, arthralgia, nausea, abdominal pain, vomiting, jaundice, and diarrhoea. Following consent by volunteer, a questionnaire was used to collect data, including socio-demographic details and some clinical signs. A 5 mL sample of whole blood was collected in EDTA

**Table 1. Primers and probes used to detect and serotype dengue virus by real time RT-PCR.**

| Virus | Primers | Sequences 5' → 3' | Gene |
|---|---|---|---|
| All Dengue viruses | DENV-All-F | AGGACYAGAGGTTAGAGGAGA | 3' UTR |
| | DENV-All-R | CGYTCTGTGCCTGGAWTGAT | |
| | DENV-All-P | FAM-ACAGCATATTGACGCTGGGARAGACC-TAMRA | |
| Dengue virus 1 | DENV-1-F | ATACCYCCAACAGCAGGAATT | 5'UTR-C |
| | DENV-1-R | AGCATRAGGAGCATGGTCAC | |
| | DENV-1-P | FAM-TTGGCTAGATGGRGCTCATTCAAGAAGAAT-TAMRA | |
| Dengue virus 2 | DENV-2-F | TGGACCGACAAAGACAGATTCTT | 5'UTR-C |
| | DENV-2-R | CGYCCYTGCAGCATTCCAA | |
| | DENV-2-P | FAM-CGCGAGAGAAACCGCGTGTCRACTGT-TAMRA | |
| Dengue virus 3 | DENV-3-F | AAGACGGGAAAACCGTCTATCAA | 5'UTR-C |
| | DENV-3-R | TTGAGAATCTCTTCGCCAACTG | |
| | DENV-3-P | FAM-ATGCTGAAACGCGTGAGAAACCGTGT-TAMRA | |
| Dengue virus 4 | DENV-4-F | CCATCCCACCRACAGCAGG | 5'UTR-C |
| | DENV-4-R | CAAGATGTTCAGCATGCGGC | |
| | DENV-4-P | FAM-ATGGGGACAGTTRAAGAAAAAYAAGGCCAT-TAMRA | |

Adapted from Leparc-Goffart et al., [17]

vacutainer tubes from the median cubital vein and centrifuged at 2000 rpm for 10min to obtain both the plasma and the buffy coat which were store at -20˚C for further analysis. Additionally, samples were screened for malaria using either a rapid diagnostic test or whole blood thick smear.

## Real time RT-PCR for dengue screening and serotype classification

Total RNA was extracted from plasma using the QIAamp viral RNA mini kit, as per the manufacturer's recommendations (Qiagen, Hilden, Germany). The extracted RNA was retrotranscribed into cDNA using the High-capacity cDNA reverse transcription kit (Applied Biosystems, Foster City, California, USA) according to the manufacturer. Then 5 μL of cDNA was amplified using the kit TaqMan Universal PCR Master Mix (Applied Biosystems, Foster City, California, USA) with primers published elsewhere [17] and targeting a 107 bp fragment in the 3'UTR to screen for infection by either of the four dengue serotypes. The reaction was pre-amplified for 02 min at 50˚C followed by 10min at 94˚C then the amplification and fluorescence quantification were done during 45 cycles for 15 sec at 94˚C and 1 min at 60˚C. Subsequently, a dengue serotype-specific real time RT-PCR was performed to determine the serotype of samples which gave a positive result during the above-mentioned RT-PCR using four different sets of primers and probes specific to the 5' UTR and capsid following the same protocol described above [17] (Table 1). All the real time RT-PCR experiments were done on the Stratagene Mx3005P qPCR machine (Agilent Technologies, Santa Clara, California, USA).

## Amplification of the envelope gene, Sanger sequencing and phylogenetic analysis

The envelope gene was amplified using three sets of primers as previously described [18] with the Kapa Taq DNA polymerase kit (Kapa Biosystems, Wilmington, Massachusetts, USA). Briefly, 5 μL of cDNA was added to a reaction mixture containing 400nM of each primer, 1.25 mM of $MgCl_2$, and 80 μM of dNTPs. The amplification was performed by incubation for 3 min at 95˚C followed by 45 cycles of 30 sec at 95˚C, 30 sec at 55˚C and 1 min at 72˚C and a

final amplification for 10 min at 72˚C on the Applied Biosystems 2720 Thermal Cycler (Applied Biosystems, Foster City, California, USA). The amplicons were visualized on a 1% agarose gel electrophorese, purified using ExoSAP-IT (Applied Biosystems, Foster City, California, USA) then directly sequenced using the Sanger technology through Microsynth SeqLab (Göttingen, Germany). The sequences obtained, one for DENV-1, three for DENV-2 and five for DENV-3, were aligned to references retrieved from GenBank and manually edited using the MAFFT algorithm from Unipro Ugene V36.0 [19]. Then, the best fitting DNA evolution model for the phylogenetic inference was determined using Mega X [20]. Hence the phylogenetic trees were built using the Maximum likelihood under the Tamura-Nei 93 (TN-93) substitution model with a discrete Gamma distribution (+G) and assuming that a certain fraction of sites are evolutionarily invariable (+I). The trees were built with 1000 bootstrap iterations and annotated using FigTree V1.4.4 (http://tree.bio.ed.ac.uk/software/figtree/).

### Statistical analysis

Pair-wise comparisons were made between dengue-positive and dengue-negative cases, using either the Chi-square ($\chi^2$), the Fischer exact test or the student t-test with significance at p-value $< 0.05$ for qualitative and quantitative variables. Independent variables were tested for possible association with dengue positivity and expressed as odds rations (OR) with 95% confidence intervals. All statistical analyses were performed using R version 4.00 (The R Foundation for Statistical Computing, https://www.r-project.org/).

## Results

From July to December 2020, 320 individuals were recruited in four public hospitals in Douala. The mean age of this population was 29 ±17.4 years, with a sex ratio M/F of 0.8. Participants comprised 16.87% below 15 years. The most common clinical signs found in our cohort were headache (63.7%), asthenia (58.7%) and joint pain (47.8%); jaundice and rash were reported from 16 and 4 patients respectively, out of the 320 individuals included in the study (Table 2). Malaria infections were detected in 227 (71%) cases using both RDT and thick smear.

Dengue virus was found in 12.8% (41/320) of participants who lived in 12 of the central and peripheral neighbourhoods in the city (Fig 1), with the highest number from Newbell, a densely populated settlement in the central ward of the city. Co-infections of dengue with malaria were seen in 56% of dengue positive cases. Three dengue serotypes were found, DENV-1, DENV-2, and DENV-3 in patients aged from 3 years to 75 years, but significantly more were in adults (81.1%) than in children below 15 years old (18.9%; $\chi^2 = 21.84$, df = 1), (Table 3). DENV-3 was the most prevalent serotype (68.3%), followed by DENV-2 (19.5%) and DENV-1 (4.9%), and 6 (14,6%) that could not be serotyped. Three cases of co-infection with DENV-2 and DENV-3 were found, though none exhibited clinical signs of severe dengue. Of all symptoms reported, headache (OR = 0.4, IC95% [0.2–1.0], $p = 0.04$) and jaundice (OR = 3.4, IC95% = [1.0–10.5], $p = 0.03$) were significantly associated with dengue infection (Table 2). Dengue cases were found in July, August, and September, which is the rainy season in Douala (Table 3).

### Phylogenetic analysis of the dengue serotypes

The phylogenetic analysis shows that our strain of DENV-1 belongs to genotype V (Fig 2), the American- African genotype which is a cosmopolite genotype including strains from around the world. Indeed, our DENV-1 belongs to the subset of DENV-1 found in West and Central Africa and is closely related to the Gabonese strain of 2012 (Fig 2). Interestingly, the cluster

**Table 2. General description of the dengue positive and negative cases found in the study.**

| Characteristics | Dengue positive (N = 41) | Dengue negative (N = 279) | OR (CI 95%) | p-value | Total (N = 320) |
|---|---|---|---|---|---|
| **Socio-demographic characteristics** | | | | | |
| **Age (years)** | | | | | |
| least than 05 | 1 | 29 | 0.730 (0.34–1.49) | **0.03** | 30 |
| 06 to 15 | 7 | 38 | | | 45 |
| 16–24 | 13 | 50 | | | 63 |
| 25–45 | 16 | 95 | | | 111 |
| 46–84 | 2 | 44 | | | 46 |
| NA | 2 | 23 | | | 25 |
| mean Age | 26.57±14.29 | 28.63±17.90 | 0.82* | 0.41 | 29 ±17.48 |
| Male (%) | 21 (51.22%) | 121 (43.37%) | 0.60# | 0.43 | 142 (44.38%) |
| Mean day of fever since the onset | 2.92±2.52 | 4.15±2.91 | 3.21* | **0.002** | 3.99 (2.81) |
| YF vaccination | 11 (26.83%) | 78 (27.96%) | 2.58e-30# | 1 | 89 (27.81%) |
| **Clinical manifestation** | | | | | |
| Headache | 26 (63.41%) | 178 (63.80%) | 0.45 (0.21–1.00) | **0.04** | 204 (63.75%) |
| Asthenia | 23 (56.1%) | 165 (59.14%) | 1.17 (0.45–3.65) | 0.74 | 188 (58.75%) |
| Joint pain | 18 (43.9%) | 135 (48.39%) | 0.88 (0.40–2.00) | 0.75 | 153 (47.81%) |
| Abdominal pain | 13 (31.71%) | 103 (38.15%) | 1.03 (0.49–2.15) | 0.95 | 116 (36.25%) |
| Muscular pain | 13 (31.71%) | 99 (35.48%) | 0.85 (0.38–1.92) | 0.70 | 112 (35%) |
| Nausea | 8 (19.51%) | 98 (35.12%) | 1.03 (0.48–2.15) | 0.95 | 106 (33.12%) |
| Vomiting | 13 (31.71%) | 89 (31.95%) | 1.39 (0.51–3.39) | 0.49 | 102 (31.87%) |
| Cough | 3 (7.31%) | 67 (24.01%) | 1.52 (0.66–3.38) | 0.31 | 70 (21.87%) |
| Diarrhoea | 7 (17.07%) | 36 (12.90%) | 0.72 (0.31–1.62) | 0.44 | 43 (13.44%) |
| Retro-orbital pain | 1 (2.44%) | 22 (7.88%) | 1.10 (0.24–3.57) | 0.88 | 23 (7.19%) |
| Runny nose | 0 (00) | 22 (7.88%) | 0.70 (0.11–2.60) | 0.64 | 22 (6.87%) |
| Jaundice | 3 (7.32%) | 13 (4.66%) | 3.45 (1.01–10.48) | **0.03** | 16 (5%) |
| Photophobia | 0 (0.0) | 14 (5.02%) | NA | 0.99 | 14 (4.37%) |
| Rash | 0 (0.0) | 4 (1.43%) | 2.46 (0.12–20.17) | 0.44 | 4 (1.25%) |

N sample size, NA not available, OR odd ratio, CI95% 95% confidence interval. * student t-test; # χ2 test

formed by the Gabonese and Cameroonian strains are basal to the cluster comprising the strains responsible for outbreaks in Angola in 2013, as well as those reported from the Republic of the Congo (RC) and Democratic Republic of the Congo (DRC) since 2015, through a spill-over in Japan. Moreover, although they all belong to genotype V, our strain is not closely related to DENV-1 strains found in West Africa recently, but rather to more ancient strains from Nigeria, and Côte d'Ivoire from 1968 and 1985 respectively (Fig 2). For DENV-2, the phylogenetic reconstruction has shown that the DENV-2 strains isolated during this study belongs to the genotype II, alongside with other African strains (Fig 3). Our strains seem to have evolved from a common ancestor with the strains responsible for epidemics in West Africa since 2014. DENV-3 phylogeny shows that the Cameroonian strains belong to the genotype III. Indeed, ours strains belongs to the African DENV-3 clade (Fig 4). They root between the West and the East-South African strains. Although the strains from Gabon, a neighbouring country, seem to have directly originated from the West African strains, DENV-3 from Cameroon seems to be more ancient and seems to have evolved from the East-South African strains including those observed in the Comoros, Madagascar and Djibouti in 2010 and 2011 respectively.

**Table 3. Description of the dengue positive cases found from July to September 2020 in Douala.**

| Patient | Age (years) | Gender | RT-PCR* (Ct) | Serotype | Dengue serotyping RT-PCR# (Ct) | Collection date |
|---|---|---|---|---|---|---|
| 1 | 11 | M | 30.8 | 3 | 23.35 | September 2020 |
| 2 | 44.12 | M | 26.99 | 3 | 19.43 | September 2020 |
| 3 | 20.02 | F | 33.67 | 3 | 25.66 | September 2020 |
| 4 | 16.45 | M | 31.99 | 1 | 24 | July 2020 |
| 5 | 37.54 | F | 33.97 | 3 + 2 | 34.84/27.77 | July 2020 |
| 6 | 16.90 | M | 31.71 | 1 | 40.02 | July 2020 |
| 7 | 23.86 | M | 27.84 | 3 | 24.4 | July 2020 |
| 8 | 19.59 | F | 37.46 | 2 | 33 | July 2020 |
| 9 | 35.63 | M | 34.44 | 2 | 29 | August 2020 |
| 10 | 15.89 | F | 38.76 | nd | - | August 2020 |
| 11 | 10.66 | M | 40.19 | nd | - | August 2020 |
| 12 | 28.46 | M | 34.64 | 3 | 40.17 | August 2020 |
| 13 | 25.56 | M | 30.56 | 3 | 28.81 | September 2020 |
| 14 | 36.58 | M | 25.73 | 3 | 23.48 | July 2020 |
| 15 | 31.57 | M | 30.41 | 3 | 23.26 | July 2020 |
| 16 | 29.98 | F | 35.69 | 3 | 27.79 | September 2020 |
| 17 | 29.58 | F | 38.52 | nd | - | July 2020 |
| 18 | 13.58 | F | 28.92 | 2 | 20.99 | July 2020 |
| 19 | 42.60 | F | 36.97 | 3 + 2 | 26/28.78 | July 2020 |
| 20 | 38.57 | M | 31.44 | 3 | 27.99 | July 2020 |
| 21 | nd | M | 26.69 | 3 | 24.69 | July 2020 |
| 22 | 42.57 | F | 31.18 | 3 | 24.43 | July 2020 |
| 23 | nd | F | 39.43 | 2 | 34.52 | July 2020 |
| 24 | 75.62 | M | 39.89 | 3 | 31.76 | July 2020 |
| 25 | 52.61 | F | 25.87 | 3 | 19.93 | July 2020 |
| 26 | 20.24 | M | 38.33 | 2 | 21.19 | July 2020 |
| 27 | 21.50 | F | 32.95 | nd | - | August 2020 |
| 28 | 21.61 | M | 33.69 | 3 | 29.7 | August 2020 |
| 29 | 19.44 | F | 40.72 | 3 | 25.95 | August 2020 |
| 30 | 26.63 | F | 41.24 | 3 | 27.67 | August 2020 |
| 31 | 23.66 | M | 26.48 | nd | - | August 2020 |
| 32 | 33.67 | F | 40.81 | nd | - | August 2020 |
| 33 | 34.09 | F | 26.48 | 3 | 18.9 | August 2020 |
| 34 | 42.32 | M | 40.81 | 3 | 36.15 | August 2020 |
| 35 | 42.29 | M | 29.12 | 3 + 2 | 34.85/21.37 | August 2020 |
| 36 | 3.01 | F | 43.38 | 3 | 37.4 | September 2020 |
| 37 | 6.02 | M | 41.5 | 3 | 37.4 | September 2020 |
| 38 | 40.13 | F | 42.37 | 3 | 38.53 | September 2020 |
| 39 | 8.02 | M | 40.21 | 3 | 37.87 | September 2020 |
| 40 | 21.38 | M | 31.69 | 3 | 24.73 | September 2020 |
| 41 | 22 | M | 35.5 | 3 | 28.9 | September 2020 |

* Real time RT-PCR that detect all the dengue cases, independently of the serotypes by amplification of a portion of the 3'UTR.

# real time RT-PCR to identify each serotype by amplification of 5' UTR and capsid. Ct cycle threshold

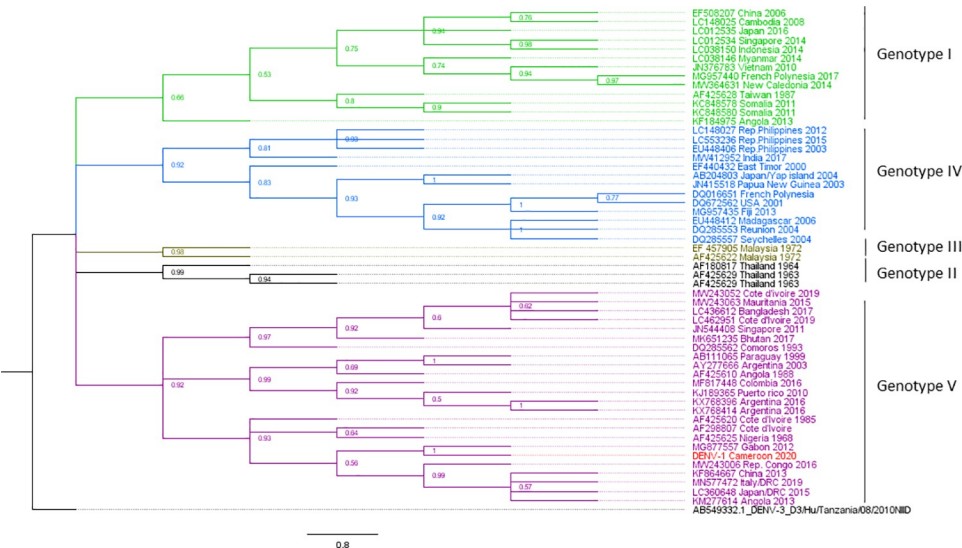

**Fig 2. Dengue virus 1 phylogenetic tree built with a 721 bp fragment of the envelope gene.** The maximum likelihood tree was built using the TN-93 +G+I model. Each DENV genotype is represented by a single colour. The sequence from this study is highlighted in red. The tree node value represented the bootstrap expressed in decimal. Dengue virus 3 from Tanzania was used as outgroup. DENV-1 from Cameroon formed a cluster with the Gabonese trains within the genotype V.

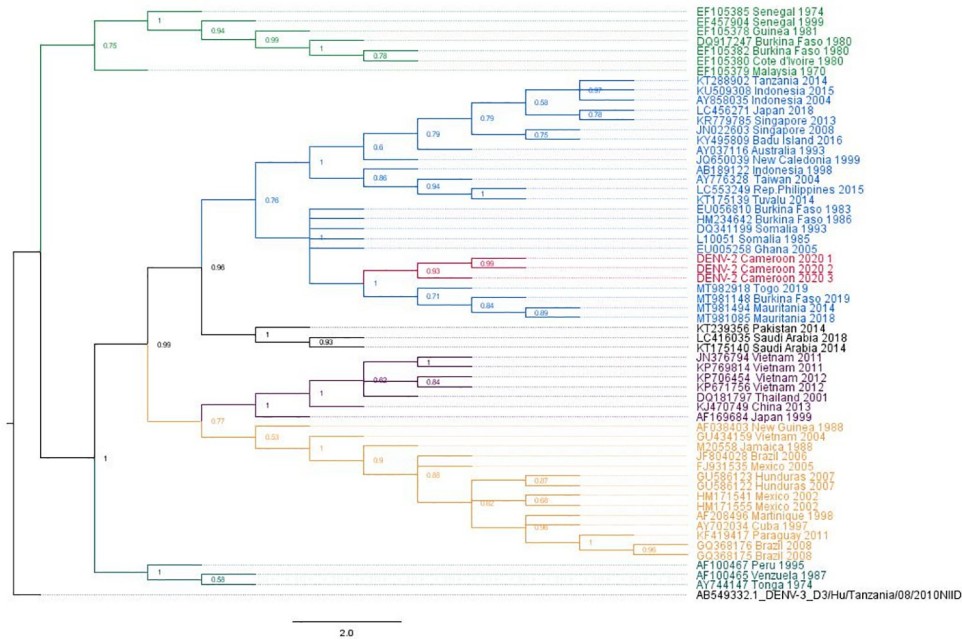

**Fig 3. Dengue virus 2 phylogenetic tree built with a 1387 bp fragment of the envelope gene.** The maximum likelihood tree was built using the TN-93 +G+I model. Each DENV genotype is represented by a single colour. The three sequences from this study are highlighted in red. The tree node value represented the bootstrap expressed in decimal. Dengue virus 3 from Tanzania was used as outgroup. DENV2 strains from Cameroon belong to genotype II along other African strains. It forms a sister clade with those responsible of the recent outbreaks in West Africa.

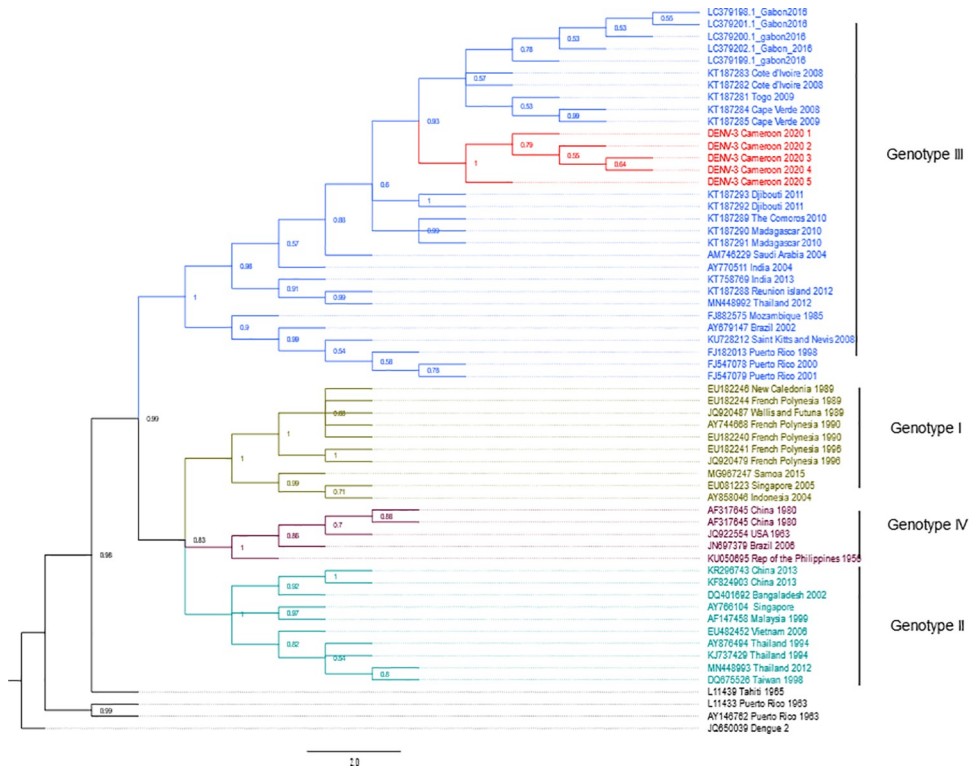

**Fig 4. Dengue virus 3 phylogenetic tree built with a 1026 bp fragment of the envelope gene.** The maximum likelihood tree was built using the TN-93 +G+I model. Each DENV genotype is represented by a single colour. The five sequences from this study are highlighted in red. The tree node value represented the bootstrap expressed in decimal. Dengue virus 4 from Singapore was used as outgroup. DENV-3 from Cameroon belongs to the genotype III. They share a common ancestor with the viruses active in West Africa since 2008.

## Discussion

To gauge the extent of the contemporary dengue threat to Cameroon, we screened for dengue infection in outpatients at four public hospitals in the city of Douala, between July and December 2020. We found a 12.8% infection rate by PCR in 320 febrile individuals examined, and three co-circulating dengue serotypes were detected. All infections were found in the wet season (July-September), predominantly in adults (81.1%).

Our results are comparable to those reported by Abe *et al.* from the same period in Gabon, where dengue infection was mainly found in adults during the rainy season [21]. This high infection rate during the wet season corresponds with the rise in abundance of the vector breeding habitats in Douala in that season [2,15,22]. Co-infections of dengue and malaria have been reported previously from Cameroon and elsewhere in Africa [7,12,13], at levels similar to the 55.6% found here. These levels of co-infection with malaria and dengue may reflect the low socio-economic status of the affected population since transmission of both pathogens by their respective vectors is favoured by poor housing quality, inadequate sanitation and poverty [23,24]. In Africa, where malaria transmission rates far exceed anywhere else worldwide, such co-infections will be responsible for substantial under diagnosis and underreporting of arboviral infections throughout the continent[25].

Three dengue virus serotypes were found during this surveillance study: DENV-1 genotype V, DENV-2 genotype II and DENV-3 genotype III. The characterisation of three dengue serotypes during the same period in one city raises a concern about severe forms of the disease,

namely dengue haemorrhagic fever (DHF) and dengue shock syndrome (DSS) through the development of antibody-dependent enhancement (ADE) in secondarily infected individuals, especially since DENV-1 and DENV-2 are already known in Douala [11,14,26,27]. Apart from jaundice, we found no clinical signs of external haemorrhage in this study. However, as we did not carry out any hematologic tests which could have informed on any eventual blood dysregulation in dengue infected individuals, nor any follow up of the patients, we cannot draw conclusions on the severity of these cases. *Aedes aegypti* and *Ae. albopictus* mosquitoes from Douala (Cameroon) have been proven to be efficient to transmit DENV-2 [16], however comparable data are not available for DENV-3 which hampers the comprehension of DENV-3 spreading in the country.

Additionally, we have found three cases of co-infections by DENV-2 and DENV-3 in individuals aged 37 and 42 years old. No specific clinical signs have been associated with those co-infections. Co-infection of individuals by numerous dengue serotypes have been recurrently reported in patients and *Aedes* mosquitoes, mostly during multi-serotype outbreaks, driven by the frequency of the circulating serotypes [28,29]. However, the relevance of such events for disease severity and on the virus life cycle has not been well established. Senaratne and colleagues have demonstrated that co-infections by two or more dengue serotypes have no incidence on disease severity since the clinical presentation, the white blood cell count and viraemia were not different from mono-infected individuals [30]. Yet, neurologic impairments have been described in children co-infected by DENV-1, -2 and -3 in Brazil [31]. More efforts should therefore be made to understand why some co-infections and the genotypes involved, result in severe cases and the situation in Douala offers opportunities for such studies in an African context.

DENV-3 genotype III found in this study was closely related to strains that have circulated in West Africa, in 2008–2009, mainly Côte d'Ivoire, Togo and Cape Verde. DENV-3 has been anecdotally described in Central Africa, Cameroon in 2006 and Gabon in 2010 [8,21,32]. DENV-3 genotype III, which originated from the Indian subcontinent, was first described in Africa in the 1980s in Mozambique and was mostly active in the eastern parts of the continent [33]. Indeed, it has been extensively described on numerous occasions in Djibouti, Sudan, Kenya [33,34,35]. However, the epidemiology of the virus has changed, as it is now recurrently found in West Africa, notably during epidemics but also from returning travellers [36,37,38,39]. The global expansion of DENV-3 genotype III is ongoing and includes Central Africa. In fact, our study pinpoints for the second time the burden of DENV-3 in Central Africa after it was first reported in Gabon during the 2016–2017. The only serotype responsible for dengue fever at the time in Gabon, was DENV-3 genotype III, which accounted for up to 8% of the febrile cases. In Douala, we found an even greater prevalence of DENV-3, however with a strain more ancient than those from West Africa and Gabon, a finding which heads to support a greater settlement of this virus in the sub-region. Indeed, one can assume the circulation of many DENV3 genotype III clades in Central Africa, one which has evolved from East African strains as seen in Cameroon and the second which has newly been introduced from West Africa, as exemplified in Gabon. DENV-3 has been postulated to be responsible for more severe manifestations of dengue disease through some meta-analysis of the disease severity, but also because the first cases of DHF have appeared with the description of distinct genetic shifts in the DENV-3 genotype III genome [33,40,41]. The establishment of DENV-3 genotype III in Cameroon and Central Africa therefore raises some public health concerns.

Interestingly, DENV-2 was the second most abundant dengue serotype found while we only got two cases of DENV-1. In a previous study in Cameroon, DENV-1 was the only serotype found among febrile patients, even in Douala [14,42,43], however, at variable prevalence. Seroprevalence studies have shown that 72% of individuals in Douala had neutralising

antibodies against DENV-2, confirming that DENV-2 was active in the city sometime in the past [11]. Hence, dengue epidemiology in Douala has shifted, with DENV-3 as the current major serotype alongside a baseline activity of DENV-2 and DENV-1. Similarly, before the finding of DENV-3 in Gabon by Abe and colleagues [21], the lately active dengue serotype in Central Africa was DENV-1, recorded from DRC [44,45,46]. In fact, the long-term persistence of a dengue lineage in a specific area has been demonstrated to be limited, due to lineage introduction, shift or replacement [47]. The introduction of new genotypes or serotypes in a region may cause bottlenecks and displacement of previously dominant serotypes, although co-circulation with the latter may occur [48,49]. Such replacement events have been well documented during major epidemics, in Peru and Malaysia and are perhaps due to a heterologous primed populations exposed to the new lineages [50,51] as well as pre-existing population immunity; but they may result in a greater number of symptomatic and severe cases [47,50,51]. Hence, if such an event is happening in Douala or Central Africa, this could lead to an increase in DENV-3 cases and severity.

Nevertheless, our study has some limitations which preclude us to conclude on some questions regarding dengue in Douala. Indeed, we have not characterised the disease at the clinical level which could have provide insights into the severity of the cases observed and hence the outcome of the disease. Additionally, we have only analysed a portion of the envelope gene of some strains. The whole genome analysis of all the viral strains found would have provide more knowledge on genetic diversity of the different lineages present.

## Conclusion

These results highlight the importance of monitoring emerging and re-emerging viruses in urban areas of Africa, especially the large high-density cities. This routine arbovirus survey has revealed dengue infections as a significant cause of acute febrile disease cases in public hospitals in Douala. We have confirmed the presence of three serotypes, DENV-1, DENV-2, and DENV-3, and of co-infections with DENV-2 and DENV-3. The presence of multiple serotypes is alarming as it demonstrates the increased likelihood of severe dengue. It is of sufficient concern to warrant further epidemiological studies and the development of appropriate dengue outbreak prevention and response plans in Cameroon and elsewhere in the region.

## Author Contributions

**Conceptualization:** Huguette Simo Tchetgna, Basile Kamgang, Philip J. McCall, Charles S. Wondji.

**Data curation:** Huguette Simo Tchetgna, Francine Sado Yousseu, Armel Tedjou.

**Formal analysis:** Huguette Simo Tchetgna, Armel Tedjou.

**Funding acquisition:** Huguette Simo Tchetgna, Philip J. McCall, Charles S. Wondji.

**Investigation:** Huguette Simo Tchetgna, Francine Sado Yousseu.

**Methodology:** Huguette Simo Tchetgna, Francine Sado Yousseu, Basile Kamgang.

**Project administration:** Philip J. McCall, Charles S. Wondji.

**Resources:** Huguette Simo Tchetgna, Basile Kamgang, Philip J. McCall, Charles S. Wondji.

**Software:** Huguette Simo Tchetgna, Armel Tedjou.

**Supervision:** Basile Kamgang, Philip J. McCall, Charles S. Wondji.

**Validation:** Huguette Simo Tchetgna, Francine Sado Yousseu, Basile Kamgang, Armel Tedjou.

**Visualization:** Huguette Simo Tchetgna, Francine Sado Yousseu, Basile Kamgang, Armel Tedjou, Philip J. McCall, Charles S. Wondji.

**Writing – original draft:** Huguette Simo Tchetgna.

**Writing – review & editing:** Huguette Simo Tchetgna, Francine Sado Yousseu, Basile Kamgang, Armel Tedjou, Philip J. McCall, Charles S. Wondji.

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
