## [Decision Letter · Decision Letter 0]

20 Jul 2021

Dear Dr SIMO TCHETGNA,

Thank you very much for submitting your manuscript "Concurrent circulation of dengue serotype 1, 2 and 3 among acute febrile patients in Cameroon" for consideration at PLOS Neglected Tropical Diseases. As with all papers reviewed by the journal, your manuscript was reviewed by members of the editorial board and by several independent reviewers. In light of the reviews (below this email), we would like to invite the resubmission of a significantly-revised version that takes into account the reviewers' comments. 

We cannot make any decision about publication until we have seen the revised manuscript and your response to the reviewers' comments. Your revised manuscript is also likely to be sent to reviewers for further evaluation.

Sincerely,

Marilia Sá Carvalho

Associate Editor

Amy Morrison

Deputy Editor

Reviewer's Responses to Questions

**Key Review Criteria Required for Acceptance?**

**Methods**

-Are the objectives of the study clearly articulated with a clear testable hypothesis stated?

-Is the study design appropriate to address the stated objectives?

-Is the population clearly described and appropriate for the hypothesis being tested?

-Is the sample size sufficient to ensure adequate power to address the hypothesis being tested?

-Were correct statistical analysis used to support conclusions?

-Are there concerns about ethical or regulatory requirements being met?

Reviewer #1: The study aims to characterize the contribution of DENV to acute febrile illness in Douala, Cameroon. The authors obtained 320 samples from symptomatic patients and were able to detect DENV in 41 samples. The positive samples were serotyped by RT-qPCR and subsequently submitted to envelope gene sequencing. Maximum likelihood analysis was performed to identify the genotypes of DENV1, DENV2 and DENV3 circulating in Duoala, Cameroon. The authors concluded that the simultaneous occurrence of three DENV serotypes in Douala reveals that the virus is a serious public health threat for Cameroon.

Reviewer #2: The objectives are clear and well delineated for the study. The design is appropriate, however the sample size was low to compose four public hospitals. Even being from a poor country with a lack of surveillance. The low number of samples that were sequenced are unrepresentative. It is known that DENV1 is well established in the Aedes aegypti mosquito on several continents. Currently, dengue viruses have originated lineages within a genotype and this is very important to show the variability, especially for dengue viruses 1 and 2. It is important to emphasize that there are already many methodologies for obtaining low-cost and sensitive sequences that allow a broad analysis.

The target population is in agreement with the proposed objectives. Statistical tests are adequate, as well as molecular analysis. Ethical aspects are reported in the text.

Reviewer #3: (No Response)

**Results**

-Does the analysis presented match the analysis plan?

-Are the results clearly and completely presented?

-Are the figures (Tables, Images) of sufficient quality for clarity?

Reviewer #1: Some inconsistencies were observed in Results section; therefore, authors should clarify or rectify their statement in the text:

Table 2: in the age column, the sum of positive dengue patients is not 41:

- 6 to 15 age line: the sum of positive and negative dengue cases is 48, but only 7 positive and 40 negative cases are listed

- 25 to 45 age line: the sum of positive and negative dengue cases is 103, but only 16 positive and 86 negative cases are listed

Lines 159-160: the number of dengue and malaria co-infections is inconsistent: 176 cases were co-infections with malaria, but only 41 patients tested positive for dengue. Is that accurate?

Lines 177, 179, 185, 187 and 190: the names of the figures should be changed to Figure 2, 3 and 4 instead of Figure 2a, 2b and 2c. 

Figure 2, Figure 3 and Figure 4: ML trees were inferred with a number of dengue sequences different from the mentioned number of sequences generated in this study, for example: the authors generated 2 DENV1 sequences, but the ML tree shows only 1 sequence. The same was observed for DENV2 and DENV3. The authors should include all sequences generated in this study for phylogenetic analysis or make it clear why only some sequences were used.

Line 193: The authors mentioned that DENV3 strains from Cameroon might have evolved from South Africa strains. For clear understanding, the authors should provide the names of the countries they are referring to.

Line 226: the ages of the patients co-infected with DENV2 and DENV3 are inconsistent with the information presented on f the Table 3.

Reviewer #2: The study proposed to determine the DENV serotypes and genotypes of dengue cases in four public hospitals, however a low number of positives was obtained by molecular biology. With regard to genetic sequencing, few sample sequences were obtained, this may be related to the sequencing methodology for genomic analysis used to determine the genotypes. There is an important reference in the study that malaria was found in more than 70% of cases and people are co-infected with two denv serotypes, such as denv2 and denv3.

The results are clear, with a lot of clinical information, which demonstrate good accuracy in data collection. This is relevant epidemiological information, as there is little data of this type in the study area, as endemic malaria makes it difficult to detect dengue cases. Studies of this type should be strengthened, as the clinical manifestations of dengue have caused serious conditions, such as dengue hemorrhagic fever and neurological alterations.

Figures are of good quality, but the table 2 needs improvement in presentation, I suggest a subdivision within it, to separate population demographic characteristics from clinical manifestations.

Reviewer #3: (No Response)

**Conclusions**

-Are the conclusions supported by the data presented?

-Are the limitations of analysis clearly described?

-Do the authors discuss how these data can be helpful to advance our understanding of the topic under study?

-Is public health relevance addressed?

Reviewer #1: The conclusions provided by the authors are sounded, but little is discussed about the limitations of the study. The authors discussed that their results are relevant for public health in Africa and how the monitoring of virus could help preventing dengue outbreaks.

Reviewer #2: The conclusion reports on the importance of arbovirus surveillance in urban areas in Africa, which is really important for public health. To improve viral surveillance, it is necessary to change the study algorithm, in order to have a closer phylogenetic profile of the genotypes, a greater number of sequences must be obtained, including using more sensitive methods, such as next-generation sequencing or at least larger partial sequences by Sanger.

At no time were the limitations of the study reported, such as difficulties with methodologies. Phylogenetic analyzes are in agreement and provided important and reliable information, but for a more detailed study of genotypes it is necessary to increase the fragment size, as well as the number of sequences.

Reviewer #3: (No Response)

**Editorial and Data Presentation Modifications?**

Reviewer #1: (No Response)

Reviewer #2: Minor Revision: Review table 2, with division of demographic information from clinics.

Reviewer #3: (No Response)

**Summary and General Comments**

Reviewer #1: Some inconsistencies were observed in Results section and related tables; therefore, authors should clarify or rectify their statement in the text.

Reviewer #2: Strength of the study:

1- Contribute to the knowledge of the situation of arboviruses in urban areas in Africa;

2- Identify circulating serotypes and genotypes;

3- Propose DENV surveillance in the study area, as three circulating serotypes were detected.

Reviewer #3: The article proposed by SIMO TCHETGNA et al., acute febrile patients were presenting at hospitals between July and December 2020 for monitoring the circulation and co-circulation of DENV in Douala, Cameroon. Still, some details in the study queue should be taken into account:

The quantity of dataset samples is not representative of the phylogenetic inference analyses. I recommend the authors check the phylogenetic signal of the dataset before carrying out the phylogenetic reconstructions, considering that it is only the envelope region of the virus, requiring a more extensive set of data to represent the cohesive statements of the cohesive statements the analyses. It would be interesting for the authors to check the introduction of serotypes. It is not common to have three dengue serotypes circulating in a small region of a country. What were the criteria used for the sequencing of the samples, considering that of the 41 positives, only nine samples were sequenced, only 21% were sequenced? Was there sequencing of co-infection samples between serotypes 2 and 3? I believe that the results need to be better worked to accept the article in the newspaper, so I don't recommend the article in its format.

PLOS authors have the option to publish the peer review history of their article (what does this mean?). If published, this will include your full peer review and any attached files.

Reviewer #1: No

Reviewer #2: No

Reviewer #3: No
---

## [Editor Report · Decision Letter 1]

28 Sep 2021

Dear Dr SIMO TCHETGNA,

We are pleased to inform you that your manuscript 'Concurrent circulation of dengue serotype 1, 2 and 3 among acute febrile patients in Cameroon' has been provisionally accepted for publication in PLOS Neglected Tropical Diseases.

Best regards,

Marilia Sá Carvalho

Associate Editor

Amy Morrison

Deputy Editor

---

## [Editor Report · Acceptance letter]

21 Oct 2021

Dear Dr SIMO TCHETGNA,

We are delighted to inform you that your manuscript, "Concurrent circulation of dengue serotype 1, 2 and 3 among acute febrile patients in Cameroon," has been formally accepted for publication in PLOS Neglected Tropical Diseases.

Best regards,

Shaden Kamhawi

co-Editor-in-Chief

Paul Brindley

co-Editor-in-Chief
